# Salivary Zinc and Copper Levels Are Differentially Associated with ROS Levels in Breast Cancer Patients

**DOI:** 10.3390/ijms26104784

**Published:** 2025-05-16

**Authors:** Elena I. Dyachenko, Elena A. Sarf, Lyudmila V. Bel’skaya

**Affiliations:** Biochemistry Research Laboratory, Omsk State Pedagogical University, 644099 Omsk, Russia; dyachenko.ea@gkpc.buzoo.ru (E.I.D.); sarf_ea@omgpu.ru (E.A.S.)

**Keywords:** saliva, breast cancer, copper, zinc, ROS, phenotype, antioxidant enzymes, hormones, cytokines, amino acids

## Abstract

Disruption of the balanced metabolism of copper and zinc can be both a consequence and potential cause-trigger for the occurrence of many pathological conditions including cancer. Zinc is an important cofactor of many enzymes that participate in inflammatory and redox reactions and the immune response, and refers to the components of DNA transcription factors. Copper plays an important role in processes such as cuproplasia and cuproptosis, affecting the process of cell differentiation and the epithelial–mesenchymal transition of cancer cells. In this regard, the study of changes in copper and zinc in breast cancer can provide valuable information on the metabolic features of cancer cells. In this study, we investigated the metabolic relationship between the zinc and copper levels in the saliva of patients with breast cancer and the content of reactive oxygen species, the state of the antioxidant and immune systems as well as the metabolism of the amino acids Cys, His, Met, and Arg. We also considered how the content and ratio of copper and zinc in saliva changes in patients with breast cancer depend on the state of the hormonal background and the expression of hormone receptors.

## 1. Introduction

Disturbance of the balance of zinc and copper metabolism in the human body is both the cause and consequence of a number of pathological processes. Particular attention has been paid to the study of the contents of copper and zinc in biological fluids in oncological diseases including breast cancer (BC) [1,2,3]. A disturbance in metabolism leads to an increase in the level of oxidative stress due to direct damage to the integrity of the cell and its intracellular membrane organelles, the disruption of mitochondria, formation and release of reactive oxygen species (ROS), and damage to the DNA structure. This in turn affects the state of immune protection, causes direct and indirect activation of cell division signaling pathways, and creates favorable conditions for metastasis and the epithelial–mesenchymal transition of cancer cells [4,5].

A new direction is the study of the metabolic activity of the heavy metals copper and zinc in saliva in breast cancer. Saliva is a promising diagnostic biomaterial in many aspects [6]. Mixed saliva contains a wide biochemical spectrum, adequately reflecting changes in metabolism at the systemic level in many benign and malignant pathologies [7,8,9]. The non-invasiveness of collection, simplicity of storage and transportation conditions of biomaterial as well as biological safety for medical personnel during its collection makes saliva attractive for scientific research. The large-scale study of the diagnostic capabilities of saliva will determine its direction in practical application.

The homeostasis of zinc and copper is maintained through strict regulation by protein transporters. Zinc is a cofactor of many enzymes that are responsible for the inflammatory process, oxidative reactions, the viability of the immune response, and regulation of the cell apoptosis process [10]. Copper homeostasis regulates numerous reactions ranging from cuproplasia (copper-dependent cell growth and proliferation) [11], cuproptosis [12], and neuronal activity [13] and ends with cell differentiation and the epithelial–mesenchymal transition of cancer cells [14,15,16]. The disruption of copper metabolism can lead to tumorigenic activity such as angiogenesis, metastasis, and cell proliferation [17]. Copper and zinc are components of copper-zinc superoxide dismutase (CuZnSOD) and numerous transcription factors [17]. Disruption of the balance of the zinc and copper ratio can lead to the development of oxidative stress and disruption of the antioxidant functions of enzymes [18].

There are 24 known zinc importers, Zrt- and Irt-related proteins (Zip), and about 10 exporters, which are Zn transporters (ZnT). The activity of Zip and ZnT is affected by sex hormones [19]. Outside the cell, zinc can bind to metalloproteinases, albumin, or individual amino acids. The copper content in the human body is determined by the alimentary intake of metal with food or by the internal redistribution of copper in the human body in accordance with biological needs. The main copper transporters are ceruloplasmin, ferroxidase, albumin, α2-macroglobulin, and to a minor extent, low-molecular compounds [20,21,22]. Copper transport, depending on its valence, is carried out by various transporters: divalent copper—divalent metal transporter 1 (DMT1) [23,24], and monovalent copper—copper transporter 1 (CTR1) [25]. Divalent copper is reduced to monovalent by duodenal cytochrome B reductase (DCYTB), a family of six-transmembrane epithelial antigen of prostate 2, 3, 4 (STEAP 2, 3, and 4) metalloproteinases, which are located on the surface of many cells including prostate tissues [26,27]. In this case, the affinity of copper with CTR1 is very high [28]. Then, CTR1 delivers copper to the antioxidant copper chaperone protein-1 (Atox1) or to the low-molecular weight copper ligand (CuL) through which copper is delivered to its target sites [29]. Monovalent copper is part of the structure of enzymes such as ceruloplasmin, dopamine beta-hydroxylase, peptidylglycine-alpha-amidating monooxygenase, tyrosinase, and lysyl oxidase [30,31]. When there is an excess of copper, it is chelated in metalloproteins [32].

We have previously shown that the copper concentration in saliva increases in breast cancer, but due to the small sample size, it was difficult to analyze the relationship with the breast cancer phenotype in detail [33]. In the present study, we additionally assessed the zinc content in saliva, which was conducted for the first time. The aim of this study was to evaluate the zinc, copper, and Cu/Zn ratio in the saliva of patients with breast cancer as well as their relationship with antioxidant system activity, hormonal, cytokine status, and free amino acid content.

## 2. Results

### 2.1. Copper and Zinc Content in Saliva in Breast Cancer

The concentration of zinc and copper in saliva in breast cancer was higher than in the control group: for zinc, it was 4.17 [3.34; 5.14] and 3.82 [2.99; 4.25] μmol/L, respectively (*p* = 0.0112), and for copper, it was 25.7 [9.47; 61.5] and 16.9 [9.7; 21.1] μmol/L, respectively (*p* = 0.0354). For fibroadenomas, an increase in the concentration of copper (19.4 [8.36; 62.0] μmol/L) and zinc (4.22 [2.81; 6.17] μmol/L) was also shown. The Cu/Zn ratio changed in the opposite way: for breast cancer, it increased (+23.7%), and for the fibroadenomas, it decreased (−17.2%) compared with the healthy control (Figure 1A).

Differences from the control group were statistically significant for zinc in the luminal A (*p* = 0.0438), luminal B(−) (*p* = 0.0181), and luminal B(+) (*p* = 0.0404) breast cancer subtypes (Figure 1B). For copper, statistically significant differences from the control group were noted in luminal B(+) (*p* = 0.0493) and non-luminal (*p* = 0.0065) breast cancer (Figure 1C). The Cu/Zn ratio was highest in the non-luminal breast cancer subtype and statistically significantly differed from other breast cancer subtypes (Figure 1D). It should be noted that for zinc, the concentration in non-luminal and TNBC was lower than in the control group, whereas for copper, regardless of the tumor phenotype, the concentration was higher than in the healthy control (Figure 1B,C).

### 2.2. Indicators of Antioxidant System Activity, Hormonal, Cytokine Status, and Content of Free Amino Acids in Saliva in Breast Cancer and Fibroadenomas

In breast tumors, the concentration of 8-OHdG in saliva was significantly higher than the control values (+25.0% for BC; +58.8% for fibroadenomas). At the same time, the maximum concentration of 8-OHdG in saliva was observed for fibroadenomas (Table 1). The level of CRP in saliva in BC was slightly increased compared with the control group (+15.00%). In patients with BC and fibroadenomas, the levels of IL-1β (*p* = 0.0123), IL-4 (*p* = 0.0001), and IL-10 (*p* = 0.0000) were significantly higher than the norm and statistically significantly different from the control values. In patients with BC, IL-18 was higher than the normal values by +5.0%, while in patients with fibroadenomas, IL-18 was slightly lower than normal by −2.1%. In the group of patients with breast cancer, the content of free estradiol (−16.9%) and progesterone (−7.5%) decreased compared with the control group. In patients with breast cancer and fibroadenomas, the catalase activity statistically significantly decreased (*p* = 0.0001), but SOD (*p* = 0.0270) and GGT (*p* = 0.0000) activity increased, and the NO (*p* = 0.0000) and MDA (*p* = 0.0007) levels increased compared with the healthy control. The content of free amino acids in saliva decreased with the exception of methionine (Met) (Table 1).

Additionally, correlation coefficients between the hormone levels and copper and zinc content in saliva were calculated. It was shown that there was a weak positive correlation between the estradiol level in saliva and copper content (*r* = 0.1851) as well as a weak negative correlation between the estradiol level and Cu/Zn ratio (*r* = −0.1842). The opposite trend showed for the progesterone level in saliva: a weak negative correlation was noted between the progesterone level and copper (*r* = −0.2082) as well as a weak positive correlation between the progesterone level and Cu/Zn ratio (r = 0.2073). When considering the correlation between the copper and zinc levels and amino acid content, only a negative correlation of medium strength was established between cysteine and the zinc level (*r* = −0.4959).

### 2.3. Antioxidant System Activity, Hormonal, Cytokine Status, and Free Amino Acid Content in Saliva in Breast Cancer Depending on Tumor Phenotype

A statistically significant increase in 8-OHdG concentration was shown in HER2-positive breast cancer, namely for the luminal B(+) (+47.1%, *p* = 0.0419) and non-luminal subtypes (+153.5%, *p* = 0.0167) (Appendix A). In patients with the luminal A and luminal B(−) breast cancer subtypes, the 8-OHdG concentrations were close to normal values (+4.2% and −7.2%, respectively) (Figure 2). For CRP, the opposite trend was observed: in patients with non-luminal and TNBC, the level of CRP in saliva was below the normal values by −18.3% and −7.2%, respectively, whereas with the positive expression of hormone receptors, the level of CRP was above normal in the case of the luminal A subtype by +136.6% (*p* = 0.0027).

The concentrations of pro-inflammatory cytokines IL-1β and IL-18 were maximally increased in the luminal A (+317.7%, *p* = 0.0004 and +33.1%), luminal B(−) (+189.9%, *p* = 0.0407 and +13.2%), and luminal B(+) breast cancer (+294.5%, *p* = 0.0092 and +22.4%) subgroups compared with the non-luminal (+176.4%, *p* = 0.0502 and −14.4%) and TNBC (+63.7% and −23.7% for IL-1β and IL-18, respectively). The concentrations of anti-inflammatory cytokines IL-4 and IL-10 increased maximally in the subgroups of HER2-positive breast cancer: luminal B(+) (+108.1%, *p* = 0.0006 and +163.6%, *p* = 0.0000) and non-luminal (+105.6%, *p* = 0.0015 and +142.7%, *p* = 0.0000). For other breast cancer subtypes, the values were lower, but statistically significant differences remained compared with the healthy controls: TNBC (+74.5%, *p* = 0.0004 and +92.0%, *p* = 0.0000), luminal A (+41.6%, *p* = 0.0069 and +102.7%, *p* = 0.0000), and luminal B(−) (+36.7%, *p* = 0.0052 and +98.7%, *p* = 0.0000).

In luminal subtypes of breast cancer, the level of free estrogen and progesterone in saliva decreased below the values characteristic of the healthy controls (Figure 2). The maximum decrease in the level of estrogen (−28.5%, *p* = 0.0350) and progesterone (−28.5%, *p* = 0.0147) was shown for the luminal A subtype of breast cancer. Only in the non-luminal subtype was the level of free hormones higher than normal: +10.2% and +8.6%. In the group of patients with TNBC, the concentration of free estrogen was reduced by −13.3%, and progesterone was increased by +7.9% compared with the control group.

Among the group of biochemical parameters, catalase activity was most significantly reduced in the luminal breast cancer subtypes: luminal A (−19.2%, *p* = 0.0201), luminal B(−) (−24.0%, *p* = 0.0036), and luminal B(+) (−34.9%, *p* = 0.0130) compared with the healthy controls. SOD activity was increased in all breast cancer subtypes, but the maximum increase in SOD activity was observed in the luminal B(+) subtype (+56.8%, *p* = 0.0099), which correlates with the maximum decrease in catalase activity in this subgroup (Figure 2). The highest NO content in saliva was shown in HER2-negative breast cancer: luminal A (+39.5%, *p* = 0.0003), luminal B(−) (+21.5%, *p* = 0.0040), and TNBC (+29.8%, *p* = 0.0176). The MDA content in the saliva of patients with the luminal A subtype did not differ from the normal values, however, significant differences were shown for luminal B(−) (+9.1%, *p* = 0.0310) and TNBC (+18.3%, *p* = 0.0115). The GGT activity was increased for all breast cancer subtypes, but to a greater extent for the hormone receptor positive subtypes: luminal A (+13.2%, *p* < 0.0001), luminal B(−) (+11.8%, *p* < 0.0001), and luminal B(+) (+12.3%, *p* < 0.0001).

Cysteine (Cys) concentration decreased in all subgroups, maximally for the non-luminal breast cancer subtype (−56.4%, *p* = 0.0296). Met content was statistically significantly increased in the luminal A (+67.3%, *p* = 0.0003) and luminal B(−) (+49.3%, *p* = 0.0245) breast cancer subtypes (Figure 2). Histidine (His) concentration in saliva was increased only in luminal A breast cancer (+24.5%) and decreased in all other breast cancer subtypes. A statistically significant decrease in the concentration of His was shown for HER2-positive breast cancer: luminal B(+) (−25.7%, *p* = 0.0155) and non-luminal (−35.7%, *p* = 0.0123). The concentration of arginine (Arg) was statistically significantly reduced in all breast cancer subtypes, with the maximum reduction shown for the luminal A subtype (Figure 2, Appendix A).

## 3. Discussion

### 3.1. Relationships Between Salivary Copper and Zinc, Cellular Damage and Immune Response Activation

According to our data, the salivary copper and zinc concentrations were closely associated with the phenotype of breast cancer. Thus, the zinc content increased for subtypes with a positive expression of hormone receptors (luminal subtypes), while the copper content increased with HER2-positive breast cancer (Figure 1).

It is known that zinc is a direct activator of the immune system [34] and is necessary for the activation of DNA repair enzymes and maintaining cell division activity [35]. Luminal A and luminal B(−) subtypes of breast cancer have a less aggressive course, in which the body’s immune defense is activated. For these subtypes, we observed an increased content of C-reactive protein (CRP), pro-inflammatory cytokines IL-1β and IL-18, and a low level of anti-inflammatory cytokines IL-4 and IL-10 (Figure 2). Such a change in cytokine activity is due to adequate hyper activation of the immune system to the ongoing pathological process, where the main goal is its suppression. The immune system’s reverse reaction was observed in non-luminal and TNBC, in which the contents of CRP, IL-1β, and IL-18 were significantly lower compared with luminal A and luminal B(−), while the levels of IL-4 and IL-10 were maximally high. Non-luminal and TNBC are among the most aggressive subtypes of breast cancer, in which cancer cells escape from immune defense (Figure 3).

Salivary copper levels were elevated in all molecular biological subtypes of breast cancer compared with the control group and fibroadenomas. A statistically significant increase in salivary copper was noted in the HER2-positive (luminal B(+) and non-luminal) breast cancer subtypes. These subtypes are characterized by high cell proliferation, cell DNA damage, and a suppressed immune response [36]. We assessed the cell proliferation activity using the Ki-67 proliferative activity index and the consumption of the most important amino acids that trigger the mammalian target of rapamycin (mTOR) pathway, which stimulates the consumption of amino acids by cancer cells for their division and growth. These amino acids include Leu, Ile, Val, Arg, and Gln. We have discussed, in detail, in our previous studies on how these amino acids change depending on the molecular biological subtype of breast cancer [37,38].

A high copper content can cause DNA damage, the suppression of antioxidant enzymes, and directly initiate cell death by cuproptosis. When cells are overloaded with copper, mitochondrial respiration regulation is disrupted [39]. Another option that leads to cell death is copper binding to lipoylated components of tricarboxylic acid (TCA), leading to mitochondrial dysfunction [39]. Active division of cancer cells together with high ROS content leads to damage of the integral structure of DNA. It is known that guanine, which is a part of DNA nucleotides, is one of the most vulnerable to oxidative damage of DNA nitrogenous bases. As a result of the oxidative damage of DNA, ROS affect 2′-deoxyguanosine, which leads to the formation of 8-hydroxydeoxyguanosine (8-OHdG). If the body’s defense mechanisms are adequate, 8-OHdG is removed by the enzyme human 8-oxoguanine DNA glycosylase 1 (hOGG1) and enters extracellular fluids in a free state [40]. According to our results, the content of 8-OHdG in saliva was higher in breast cancer than in the saliva of the healthy controls (Table 1). The highest value was noted in the luminal B(+) and non-luminal breast cancer subtypes (Figure 2). Some sources in the literature have mentioned that low 8-OHdG levels in the blood are a poor prognosis for breast cancer [41]. In our experiment, an increase in the 8-OHdG levels was directly correlated with a deterioration in the antioxidant defense system of patients with the most aggressive breast cancer subtypes and a high copper level in saliva. With luminal A and luminal B(−), where the zinc content in saliva was significantly higher compared with the control group and other molecular biological subtypes of breast cancer, a slight increase in the 8-OHdG levels was noted, and with luminal B(−), the values were slightly below normal. We associate this with the fact that with these breast cancer subtypes, the antioxidant defense system is in an active state and ROS are neutralized.

### 3.2. Relationship Between Androgen Hormones and Copper and Zinc Metabolism

Some studies have discussed the mutual influence of zinc and copper on the sensitivity of receptors to estrogen and progesterone as well as the effect of hormones on the metabolism of copper and zinc. With regard to zinc, it has been shown that estrogen potentiates the activity of zinc importers and blocks the activity of exporters [42,43]. Therefore, estrogen deficiency increases zinc excretion, while normal or high estrogen levels enhance intracellular zinc uptake. Estrogen has an inverse effect on copper levels. Estrogen increases the formation of ceruloplasmin and free copper. The exact mechanisms by which estrogen affects copper levels have not yet been described [44]. Thus, estrogen can directly stimulate ceruloplasmin production [45] and the upregulation of copper-transporting P-type ATPase (ATP7A) mRNA, resulting in increased intestinal copper absorption, which leads to an increase in copper concentrations in biological fluids [46,47]. There is no recent information on how progesterone affects zinc and copper metabolism. It is only known that the use of oral contraceptives with progesterone normalizes zinc and copper levels, reducing their elevated amounts in free form [48,49]. Investigation of the mechanism of action of the hormones estrogen and progesterone on zinc and copper metabolism would be very valuable. Several studies have investigated the effect of copper in copper-containing intrauterine devices (Cu-IUDs) on the sensitivity of receptors to estrogen and progesterone [50,51]. It was shown that copper is able to competitively bind to estrogen and progesterone receptors, which led to a decrease in the sensitivity of these receptors to hormones [52]. Figure 3 schematically shows the relationship between copper, zinc, free hormone levels, hormone receptor expression, and transmembrane zinc transporters.

In our work, we observed a relationship between the levels of zinc and copper in saliva and the presence or absence of the expression of hormonal receptors of estrogen and progesterone. Thus, in patients with luminal subtypes of breast cancer, the level of free estrogen and progesterone was reduced, but the level of zinc in saliva was increased. As above-mentioned, low estrogen levels increase zinc excretion. Zinc increases the sensitivity of estrogen and progesterone receptors. Copper concentrations in the group of patients with the luminal A and luminal B(−) subtypes were slightly higher than normal, while with luminal B(+), the copper level was comparable to the non-luminal subtype of breast cancer. Most likely, the high copper content in these subgroups characterizes the presence of oxidative stress to a greater extent than the influence of hormones.

In patients with non-luminal and TNBC, concentrations of free estrogen and progesterone were the highest, including those higher than in the healthy controls. At the same time, these patients had reduced zinc levels and increased copper levels in saliva. This is consistent with the above assumption that high estrogen levels lead to a decrease in the sensitivity of hormonal receptors and zinc excretion as well as an increase in the extracellular copper content.

Thus, changes in zinc and copper metabolism in saliva have characteristic features depending on the phenotype of breast cancer, namely on the expression of hormonal receptors. A distinctive feature of luminal B(+) from other subtypes of breast cancer may be a simultaneous increase in copper and zinc in saliva.

### 3.3. The Influence of Copper and Zinc on the Processes of Lipid Peroxidation and Enzymatic Activity of the Antioxidant System

According to our data, breast cancer is accompanied by an increase in ROS, which is directly evidenced by the content of the primary radical nitric oxide (NO) and indirectly by the copper level. Copper takes an active part in oxidation–reduction reactions [53]. Thus, in the Fenton and Haber–Weiss reaction, copper is able to react with hydrogen peroxide and form a hydroxyperoxide radical. The hydroxyperoxide radical has a high oxidative potential (2.8 V) [54], which leads to damage to the lipids of the cytoplasmic membrane as well as membrane organelles [55]. Hydrogen peroxide can directly damage the cell membranes, although its direct oxidative potential is quite weak (1.8 V) [56]. It can also dismutate into the highly reactive hydroxyperoxide [57]. Excessive hydrogen peroxide content leads to its accumulation in both the extracellular space and diffusion into the cell through the water-carrying proteins aquaporins, stimulating the formation of SOD. In turn, SOD affects the formation of hydrogen peroxide, and the higher the content and activity of SOD, the greater the production of hydrogen peroxide [58,59,60]. Another source of hydrogen peroxide and singlet oxygen, under high oxidative stress, can be the spontaneous dismutation of two superoxide anion radicals [61,62]. Thus, as soon as the process of lipid peroxidation and ROS formation is launched, this process becomes self-activating. Regulation of this process occurs through antioxidant enzymes (Figure 3). According to our data, with luminal B(+) and non-luminal subtypes of breast cancer, we saw the highest copper content, which led to the accumulation of ROS and a moderately increased content of NO in saliva. The accumulation of copper leads, as already mentioned above, to an increase in ROS, which, in turn, leads to an increase in the final products of lipid peroxidation MDA [63]. In our study, we saw the accumulation of the lipid oxidation end product MDA in all molecular biological subtypes of breast cancer compared with the healthy controls, without any specific patterns for individual breast cancer subtypes.

There are several known forms of SOD: SOD1 depends on the presence of copper and zinc and is located in the cytoplasm of cells; SOD2 depends on the presence of manganese and is located in the mitochondria; SOD3 also depends on the presence of copper and zinc and is located in the extracellular space [59]. The role of zinc in the SOD enzyme is to play a structure-forming role in the protein, while copper is responsible for the activity of the enzyme and is part of the catalytic center [59]. Thus, the activity of SOD is proportional to the copper content. Zinc can be replaced by cobalt or copper, while copper cannot be replaced by another metal [59]. This leads to the fact that SOD can only contain copper. In a recent study, this form of the enzyme was also called the copper-SOD-repeat protein (CSRP) [64]. In our study, patients with the luminal B(+) and non-luminal breast cancer subtypes had the highest copper content, which correlated with a high SOD activity. In the luminal A subtype, we observed less pronounced SOD activity and a slight increase in copper levels. Luminal B(−) and TNBC had higher SOD activity and high copper levels compared with luminal A, but were lower than in patients with the luminal B(+) and non-luminal breast cancer subtypes (Figure 2). It can be hypothesized that SOD activity in the luminal A and luminal B(−) subtypes was due to an increase in zinc concentration, whereas in the luminal B(+) and non-luminal subtypes, the activity of all-copper SOD increased. The validity of this hypothesis remains to be confirmed in future studies.

According to the literature, the level of catalase in oncological diseases, including breast cancer, decreases [65]. This is due to the presence of chronic oxidative stress in cancer, which results in chromatin remodeling. Acetylation of histone H4 leads to the opening of repressive chromatin structures and has an effect on the promoter regions of the catalase gene [66]. In the luminal B(+) and non-luminal breast cancer subtypes, we obtained the most elevated SOD activity and suppressed catalase activity. In the luminal A and luminal B(−) breast cancer subtypes, the activity of free SOD was close to the normal values, and the catalase activity was less suppressed compared with the luminal B(+) and non-luminal breast cancer subtypes. Thus, we can talk about a relatively balanced response of the body to ROS in the luminal A and luminal B(−) breast cancer subtypes.

The GGT enzyme is located on the surface of the membrane of many cells [67]. The activity of glutathione (GSH) metabolism is directly related to the activity of GGT, which is located on the surface of the cell membrane. The enzyme captures extracellular GSH, which results in the formation of the dipeptide Gly-Cys and y-Glu. The dipeptide is then cleaved by dipeptidase to Gly and Cys, which are transported by transmembrane amino acid carriers into the cell for de novo GSH synthesis. y-Glu is also transported into the cell, and under the action of 5-hydroxyproline, is converted to Glu, which is part of the first stage of GSH synthesis [68,69]. GSH is a cofactor for the formation of glutathione peroxidases (GPxs), which neutralize excess hydrogen peroxide [70]. In our study, the GGT activity was higher in patients with the luminal subtypes of breast cancer. The zinc levels were higher than the control values, and a slight increase in copper was observed, indicating the viability of the antioxidant system. It can be assumed that the ROS in these breast cancer subtypes formed in quantities that stimulated the formation of GSH, rather than leading to its depletion, as in the non-luminal and TNBC. According to our data, the more aggressive the breast cancer phenotype, the higher the ROS content and the lower the GGT activity.

In addition to the analysis of the individual zinc and copper contents, calculating the Cu/Zn ratio is informative. It is known that a high Cu/Zn ratio is unfavorable in various pathologies including breast cancer [71]. A high Cu/Zn ratio leads to the damage of cellular structures, facilitates the epithelial–mesenchymal transition of cancer cells, and promotes metastasis [71]. An imbalance toward copper accumulation and a decrease in zinc levels lead to copper triggering oxidative processes, displacing zinc from most transporter proteins that carry zinc for antioxidant enzymes. The result is a failure of the body’s antioxidant defense [72]. In our study, in patients with the luminal B(+) and non-luminal breast cancer subtypes, the Cu/Zn ratio was higher than the norm and other breast cancer subtypes. In the non-luminal breast cancer subtype, the Cu/Zn ratio was maximally high and statistically significantly different from the healthy controls. In patients with the luminal A and luminal B(−) subtypes of breast cancer, a low Cu/Zn ratio was noted due to the high levels of zinc and low levels of copper, which is logically consistent with the non-aggressive nature of these breast cancer subtypes.

For TNBC, the formation of ROS, the antioxidant enzyme activity, the copper content, and the Cu/Zn ratio occupied an intermediate position between the values for the luminal A and B(−) subtypes and HER2-positive (luminal B(+) and non-luminal) breast cancer subtypes. This is primarily due to the heterogeneity of the TNBC subgroup [73,74]. In this regard, we did not see any distinctive features in the nature of the biochemical reactions occurring in the TNBC cells.

### 3.4. Changes in the Content of Free Cys, Met, His, Arg Due to Disruption of Copper and Zinc Homeostasis

Changes in zinc and copper homeostasis are inextricably linked with changes in the immune response activity and functionality of a wide range of enzymes, including enzymes with antioxidant activity, as well as the redistribution and metabolism of amino acids [75,76]. In our study, we measured the content of Cys, Met, His, Arg in terms of both the chelating properties with respect to heavy metals (Cys, His) and the effects on the antioxidant defense system and cellular proliferation (Cys, Met, His, Arg).

From Arg, under the influence of NOS (NO synthase), citrulline (Cit) and nitric oxide (NO) are formed [77]. In addition, Arg affects cell division, ammonia removal, hormone release, and regulates the immune response by affecting T cells [78]. As for the antioxidant role of Cys, Cys is the limiting amino acid in the synthesis of GSH, the main agent of antioxidant protection [58]. Cys has chelating properties and binds copper and zinc and is contained in the metallothionein (MT) protein [79]. Zinc plays a special role in MT, since it is part of the structural center of the protein [80]. With an increase in the level of ROS formation, zinc binds to reactive particles, which leads to the oxidation of cysteine and its release from the protein along with zinc. This leads to MT polymerization, which is a signal to initiate MT synthesis. Thus, zinc in a complex with Cys in the MT protein performs antioxidant functions [81].

His, when converted to histamine by histidine decarboxylase and the cofactor pyridoxal phosphate, is also capable of forming NO [82]. Histamine is an immunomodulator and activates the formation of pro-inflammatory IL-1β and IL-18 [83]. In proteins, zinc is capable of binding to water and His, forming catalytic centers [47]. A large amount of His is contained in zinc importer proteins, Zip. In the extracellular region of Zip M1, His facilitates the capture of extracellular zinc, and the intracellular region M2 ensures the transport of zinc into the cell [84,85]. Another important property of His is its participation in cell proliferation. The breakdown of His is through intermediate reactions with the formation of tetrahydrofolate (THF). THF is necessary for the metabolism of amino acids and nucleic acids [86]. Met plays an important role in DNA methylation [87]. Through the transition first to homocysteine, and then to cystathionine, it is ultimately converted into Cys [88]. Thus, Met compensates for the deficiency of Cys. In addition, Met, like His, is capable of including THF in the formation [89].

In patients with the luminal A subtype of breast cancer, we saw the highest level of NO and a low level of Agr compared with the control group and other subtypes of breast cancer, which indicates the consumption of Agr for NO synthesis. NO is a vasodilator and activator of the immune system with the involvement of pro-inflammatory cytokines. In patients with this subtype, the level of IL-1β and IL-18 increases, and the level of anti-inflammatory cytokines IL-4 and IL-10 decreases (Figure 2). NO also belongs to ROS, which stimulates the oxidation of Cys-Zn in MT, releasing zinc and Cys. A low content of free Cys in the saliva of patients with the luminal A subtype of breast cancer with simultaneously high GGT activity indicated the activity of GSH metabolism and the viability of antioxidant protection. In the group of patients with the luminal A subtype of breast cancer, Met was higher than the normal values, indicating the compensatory replenishment of Cys deficiency, a supply of THF to actively dividing immune cells, and the methylation of their DNA. Only this group of patients had the highest His content in saliva compared with the norm and other subtypes of breast cancer, which was due to a number of reasons. Zinc, binding to His in MT proteins, enters the catalytic center. The release of zinc from MT leads to the excitation not only of Cys, but also His. In this case, there was no large consumption of His for copper neutralization and the formation of THF and NO. The luminal B(−) subtype differed from the luminal A subtype by a lower content of His and a higher content of Arg relative to other subtypes of breast cancer. We assumed that in this subtype of breast cancer, NO synthesis occurs due to His, which in this case is a metabolic feature of the luminal B(−) subtype of breast cancer.

In the group of patients with luminal B(+) breast cancer, the content of Cys and Met in saliva was significantly lower compared with the healthy group and other breast cancer subtypes. This was due to more active cell proliferation and GSH synthesis in response to increased ROS content due to the cytotoxic effect of copper. At the same time, Met, together with His, is also spent on the synthesis of THF, which is necessary for active cell division. As is known, luminal B(+), due to the positive expression of HER2, refers to aggressive breast cancer subtypes with progressive cell division. Luminal B(+) is characterized by a high copper content and a high Cu/Zn ratio, which leads to the damage of cellular structures and the formation of ROS. Free Cys and His bind to excess copper. There was a reduced Arg content compared with the control group, but a relatively high content compared with other breast cancer subtypes as less was spent on NO synthesis, which is what we observed in our study.

In patients with non-luminal and TNBC, we observed the maximum decrease in Cys concentration and a compensatory increase in Met. The GGT activity in these groups was reduced, especially in the group of patients with the non-luminal subtype of breast cancer. Cys, in this case, was not spent on GSH synthesis, since we observed reduced GGT activity. We assumed that, as in the case of the luminal B(+) subtype, free Cys chelates copper and is also spent on the increased metabolic needs of actively dividing cancer cells. The Arg content in saliva did not differ from the luminal B(+) group. The His level was also at low levels, as in the case of luminal B(−) and luminal B(+). Low His levels in aggressive subtypes of breast cancer are apparently associated with active cell division.

### 3.5. The Combination of Mutations in the BRCA1 Gene and Pathologically Altered Levels of Zinc and Copper as a Potential Risk for the Development of Breast Cancer

The presence of pathogenic mutations in the BRCA1 tumor suppressor gene increases the risk of developing a number of oncological diseases including breast cancer (up to 70%) [90]. Normally, BRCA1 homologous recombines DNA, eliminating its double-strand breaks, thereby maintaining genomic stability [91]. Recently, a prospective study discussed the potential risk of developing breast cancer with the synergistic effect of a mutation in the BRCA1 gene, high copper levels, and a high Cu/Zn ratio in the blood serum [90]. In addition to describing the individual properties of zinc and copper metals, which are also detailed in our study, the scientists highlighted the main mechanism of genomic stability disruption with simultaneous mutation in BRCA1 and high oxidative stress with copper accumulation and zinc reduction. The study indicated that BRCA1 indirectly regulates the level of oxidative stress through stabilization of the p53 protein, which promotes the transcription of p53-dependent genes involved in DNA repair. The BRCA1 gene also polyubiquitinates G2/M proteins that regulate the cell cycle. During polyubiquitination, BRCA1 marks aberrant G2/M for their subsequent degradation [92,93,94]. With a mutation in the BRCA1 gene, the genetic apparatus of the cell is in a vulnerable state due to reduced reparative capabilities. Increased copper levels, reduced zinc levels and, as a result, a high Cu/Zn ratio lead to the accumulation of ROS, which damage the DNA structure. The combination of increased oxidative stress, reduced ability to repair damaged DNA, lack of apoptosis of genetically damaged cells, and their further division logically leads to the risk of developing oncology including breast cancer [90]. It is important to note another study that assessed the association of low zinc levels and BRCA1 mutations with the risk of developing breast cancer and ovarian cancer. In the preliminary results of the researchers, they did not find an association between the serum zinc levels and the risk of developing cancer in BRCA1 mutation carriers [95]. Based on this, we can conclude that only a combined change in the zinc and copper levels leads to an increase in the Cu/Zn ratio and the occurrence of oxidative stress. Impaired antioxidant function of zinc and BRCA1 mutations in the absence of a trigger in ROS production due to high copper levels are insufficient to initiate the pathological process.

In our study, we investigated the changes in the zinc, copper, and Cu/Zn levels in the saliva of breast cancer patients. The results were informative and consistent with the postulates of the adverse effects of low zinc, high copper, and Cu/Zn ratios. A limitation of this study was the lack of information on patients with BRCA1 mutations. Therefore, we could not study the association between BRCA1 mutation and changes in the salivary zinc and copper levels. We plan to explore this issue in future studies. At this interim stage, we observed that TNBC patients had Cu/Zn ratios similar to those in the luminal A breast cancer patients. It is known that the most common mutations in the BRCA1 gene occur in TNBC patients [96]. At the same time, TNBC is not a homogeneous subtype of breast cancer and includes both a subgroup with the positive expression of hormone receptors of the luminal A-type breast cancer subtype and a subgroup of cells with a basal-like phenotype [74]. In future studies, it will be interesting to examine the heterogeneity of TNBC depending on BRCA1 mutations, copper and zinc levels, and the Cu/Zn ratio.

A further limitation of this study was the lack of division of breast cancer patients into pre- and postmenopausal groups. The presence of menopause may affect the state of biochemical parameters and amino acids. The study also did not discuss the state of the oral microbiome and its metabolic products in healthy volunteers and breast cancer patients. Understanding how the oral microbiome and the presence of cancer affect the biochemical parameters of saliva will allow us to more accurately explain the reasons for their changes. We also did not evaluate the state of the hematosalivary barrier of the salivary glands and the lining epithelium of the oral cavity. However, we can discuss its damage based on biochemical parameters that normally do not pass the hematosalivary barrier, or pass in minimal quantities. It would be interesting to track the morphological change in tissues with a parallel study of the biochemical parameters of saliva and the composition of the oral microbiome. Another limitation of the study also includes not taking into account the presence of comorbidities that can affect the level of copper and zinc in saliva.

## 4. Materials and Methods

### 4.1. Study Design

The study involved 178 patients with breast cancer, 134 patients with benign breast tumors (fibroadenomas), and 58 healthy volunteers. Patients were recruited from the admissions department of the Clinical Oncology Dispensary (Omsk, Russia). Healthy controls were recruited from the blood transfusion department of the Clinical Oncology Dispensary (Omsk, Russia).

Inclusion criteria: Female gender, patients aged 30–70 years, absence of any treatment, including surgery, chemotherapy or radiation, absence of signs of active infection (including purulent processes), and good oral hygiene. Exclusion criteria: Absence of histological verification of diagnosis. The study was approved at a meeting of the Ethics Committee of the Clinical Oncology Dispensary (protocol no. 15, 21 July 2016) and the Institutional Review Board (protocol no. 46-04/2, 20 March 2024).

In all patients in the main group, invasive breast carcinoma of the following stages was histologically and cytological confirmed: Stage I—32 (18.0%), Stage II—75 (42.1%), Stage III—67 (37.6%), and Stage IV—4 (2.3%). In 86 patients, no signs of metastasis to regional lymph nodes were detected (N0 – 48.3%); in 92 patients, metastases were detected in the displaced axillary lymph nodes (N1-3 – 51.7%). TNM classification was carried out in accordance with the AJCC (8th edition, 2017). Breast tumors were classified by the degree of tissue differentiation into highly and moderately differentiated (G I + II, *n* = 107) and poorly differentiated (GIII, *n* = 71). In all cases, the status of HER2, estrogen, and progesterone receptors was determined. A total of 120 patients (67.4%) had a confirmed HER2-negative status, 58 (32.6%) were HER2-positive; 59 patients (33.1%) were ER-negative status, 119 (66.9%) were ER-positive; 89 patients (50.0%) were PR-negative; and 89 (50.0%) were PR-positive. Ki-67 values less than 20% were determined in 68 patients (38.2%), and more than 20% in 110 patients (61.8%). According to molecular biological subtypes of breast cancer, the patients were distributed as follows: triple negative (TNBC)—32 (18.0%), luminal A—41 (23.0%), luminal B (HER2-negative) = luminal B(−)—47 (26.4%), luminal B (HER2-positive) = luminal B(+)—31 (17.4%), and non-luminal—27 (15.2%). No breast pathologies were detected in the volunteers of the control group during routine mammographic and ultrasound examinations.

All patients had saliva samples collected once, strictly before the start of treatment, and the copper and zinc contents, antioxidant system activity indicators, hormonal and cytokine status, and free amino acid content were determined.

### 4.2. Collection, Storage, and Pre-Treatment of Saliva Samples

Saliva was collected on an empty stomach between 8 and 10 a.m. by spitting without stimulation into sterile polypropylene centrifuge tubes with a screw cap in a volume of 2 mL. Immediately after collection, the saliva samples were centrifuged to separate cellular debris and reduce turbidity at 7000 rpm for 10 min (CLb-16, Moscow, Russia). The supernatant was transferred to Eppendorf tubes and frozen at −80 °C until used for study.

### 4.3. Determination of Copper and Zinc Content in Saliva

The content of heavy metals in saliva was determined by the fluorometric method (Fluorat-02–5M liquid analyzer, Lumex, St. Petersburg, Russia). The method for measuring copper concentration (μmol/L) was based on the reaction of the formation of fluorescent dimers of lumokupferon (CAS number: 3626-95-7, St. Petersburg, Russia) catalyzed by copper ions in a slightly alkaline medium. The method for measuring zinc concentration (μmol/L) was based on the formation of a complex compound with 8-mercaptoquinoline (CAS number: 2801-16-3, Russia) in an acetate buffer solution with pH = 4.6 – 4.9, its extraction with chloroform, and the measurement of the fluorescence intensity of the extract. The concentration of copper and zinc was calculated using a pre-built calibration graph. The detection limit of copper and zinc was 0.1 μmol/L.

### 4.4. Determination of Cytokines, Hormones, C-Reactive Protein, and 8-OH-Deoxyguanosine by ELISA

The content of cytokines (IL-1β, IL-4, IL-10, IL-18) in saliva was determined by the solid-phase enzyme immunoassay method using the Vector-Best Kit (Novosibirsk, Russia). The concentration of CRP (mU/mL) was determined by the highly sensitive enzyme immunoassay method using the Vector-Best Kit (Novosibirsk, Russia). Determination of 8-OH-deoxyguanosine (8-OhdG) was carried out by the competitive enzyme immunoassay method using the Cloud-Clone Corp Kit (pg/mL). Determination of estradiol (nmol/L) was carried out by the competitive enzyme immunoassay method using the Hema Kit (St. Petersburg, Russia), and progesterone (nmol/L) using the Vector-Best Kit (Novosibirsk, Russia). The analysis was performed on a Thermo Fisher Multiskan FC analyzer (Waltham, MA, USA). All studies were performed in accordance with the manufacturer’s instructions without changing the volumes of the reagents and samples. The calculation of concentrations in all cases was carried out according to a previously constructed calibration graph.

### 4.5. Determination of Free Amino Acids in Saliva

High performance liquid chromatography was used to determine the concentration of amino acids (Cys, Met, His, and Arg) using a 1260 Infinity II chromatograph with selected reaction monitoring detection on a 6460 Triple Quad mass spectrometer (Agilent, Santa Clara, CA, USA). The internal standard was Alanine-d4 (sc-480386, Santa Cruz Biotechnology Inc., Dallas, TX, USA). At least six samples of the amino acids in the Plasma/Urine LC-MS/MS Analysis Kit (Jasem, Turkey) were used to construct the calibration scale. Automatic integration of chromatograms was used using Quantitative Quant-my-way software (MassHunter Workstation Quantitative Analysis B.09.00, Agilent, Santa Clara, CA, USA).

### 4.6. Determination of the Biochemical Composition of Saliva

In all saliva samples, five biochemical parameters were determined using a StatFax 3300 semi-automatic biochemical analyzer (Awareness Technology, Palm City, FL, USA). In all samples, the content of malondialdehyde (MDA, μmol/L) was determined [97]. Superoxide dismutase (SOD, c.u.) activity was determined by the accumulation of the adrenaline autoxidation product by the superoxide anion radical in an alkaline medium, and the catalase activity (nkat/L) by the ability of hydrogen peroxide to form a stable colored complex with molybdenum salts. The intensity of nitric oxide (NO, μmol/L) synthesis was estimated by the content of stable products of its oxidation—nitrate ions. Gamma-glutamyl transferase (GGT, U/l) activity was determined by the kinetic method using L-gamma-glutamyl-3-carboxy-4-nitroanilide as a substrate according to Seitz-Persin.

### 4.7. Statistical Analysis

The distribution and homogeneity of variances in the groups were preliminarily checked. According to the Shapiro–Wilk test, all of the determined parameters did not correspond to the normal distribution (*p* < 0.05). The conducted test for the homogeneity of variances in the groups (Bartlett’s test) allowed us to reject the hypothesis that variances were homogeneous across groups (*p* < 0.0001). Therefore, nonparametric statistical methods were used to process the data.

Statistical analysis of the obtained data was performed using Statistica 13.3 EN software (StatSoft, Tulsa, OK, USA) through a nonparametric method using the Wilcoxon test in dependent groups and the Mann–Whitney U test in independent groups. The sample was described using the median (Me) and interquartile range in the form of the 25th and 75th percentiles [LQ; UQ]. Differences were considered statistically significant at *p* ˂ 0.05.

## 5. Conclusions

It was shown that the zinc content in saliva is higher than normal in patients with luminal subtypes of breast cancer. Zinc promotes the stabilization and reduction in DNA damage and activates the immune system and antioxidant protection, as evidenced by the low content of 8-OHdG in saliva with more prognostically favorable luminal A and luminal B(−) subtypes of breast cancer. For these subtypes, the highest concentration of pro-inflammatory cytokines IL-1β and IL-18 and GGT activity in saliva, moderate activation of SOD, and pronounced suppression of catalase activity were shown, and the level of free estrogen and progesterone was reduced. Apparently, the low level of free estrogens increased zinc excretion. Amino acids Cys, Met, His, and Arg participated in the chelation of heavy metals (Cys, His), promoted the enzymatic activity of GGT (Cys, Met), and participated in the synthesis of NO (Arg), which was both a damaging agent for cancer cells and promoted the increased permeability of blood vessels for the migration of immune cells to the site of the pathological process. Thus, zinc both activated the antioxidant and immune defense systems, and its level increased due to the activity of antioxidant and immune defense.

The copper level was increased in all molecular biological subtypes of breast cancer, but statistically significant changes were shown in HER2-positive status. In this case, in the non-luminal subtype, a high copper level was accompanied by a low zinc level in saliva, which led to a high Cu/Zn ratio. A high copper level caused an increase in the 8-OHdG content, failure of antioxidant defense with the hyper production of SOD, suppressed activity of GGT and catalase, and the activation of anti-inflammatory cytokines IL-4 and IL-10. The luminal B(+) and non-luminal subtypes of breast cancer were characterized by the greatest depletion of free amino acids, which indicates both their active consumption by cancer cells and increased expenditure on binding heavy metals. Consequently, a high copper level is associated with an increase in ROS level, high oxidative stress, and suppressed immune and antioxidant defense.

TNBC requires separate consideration, since the metabolic changes that we observed were characteristic of both more aggressive subtypes of breast cancer (luminal B(+) and non-luminal) and less aggressive ones (luminal A and luminal B(−)). This once again emphasizes the internal heterogeneity of the TNBC subtype and the expediency of its division based on its genetic, biological, metabolic, and morphological features. Furthermore, we separately highlighted the luminal B(+) subtype of breast cancer, which is characterized by a high content of zinc and copper in saliva. Another feature is the high Cu/Zn ratio, which indicates the aggressive nature of the oncological process and serves as an unfavorable prognostic indicator.

To date, there have been few studies on the changes in the metabolic composition of saliva in various oncological diseases including breast cancer. Our study showed that saliva can be a fairly informative biological material in relation to some metabolites in breast cancer. Further research in this area will form a deeper understanding of the metabolic changes occurring in the human body in oncological diseases that affect the composition of saliva as well as open new horizons for the use of saliva to analyze a wider metabolic spectrum.

## Figures and Tables

**Figure 1 ijms-26-04784-f001:**
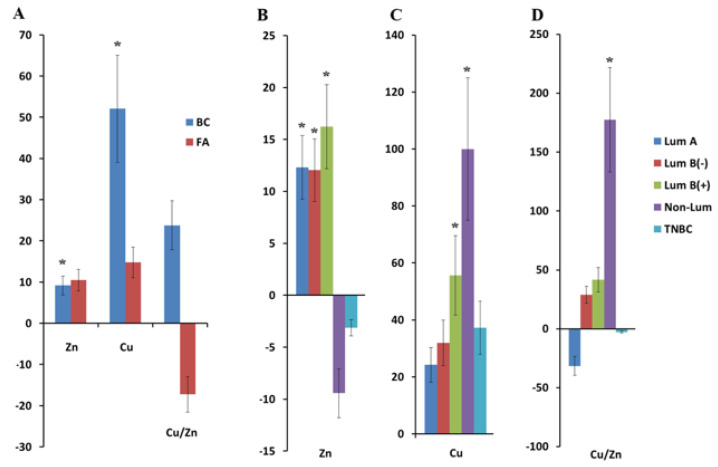
Relative change in the concentration of copper, zinc, and Cu/Zn in saliva for breast cancer and fibroadenomas compared with the healthy controls (**A**). The effect of the molecular biological subtype of breast cancer on the relative content of zinc (**B**), copper (**C**), and Cu/Zn (**D**) in saliva compared with the healthy controls. * Differences with healthy controls were statistically significant, *p* < 0.05.

**Figure 2 ijms-26-04784-f002:**
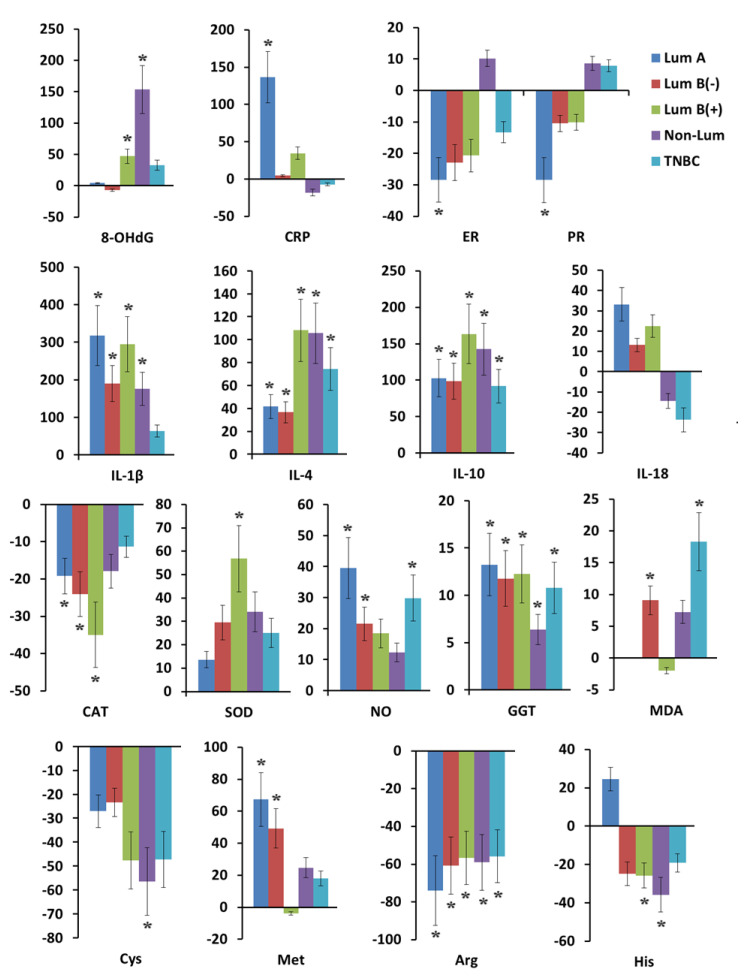
Relative change in the activity of the antioxidant system, hormonal, cytokine status, and the content of free amino acids in saliva in breast cancer depending on the tumor phenotype compared with the healthy controls, %. * Differences with healthy controls were statistically significant, *p* < 0.05. 8-OHdG—8-OH-deoxyguanosine, CRP—C-reactive protein, ER—estradiol, PR—progesterone, IL—interleukin, CAT—catalase, SOD—superoxide dismutase, NO—nitric oxide, GGT—gamma-glutamyl transferase, MDA—malondialdehyde, Cis—cysteine, Met—methionine, Arg—arginine, His—histidine.

**Figure 3 ijms-26-04784-f003:**
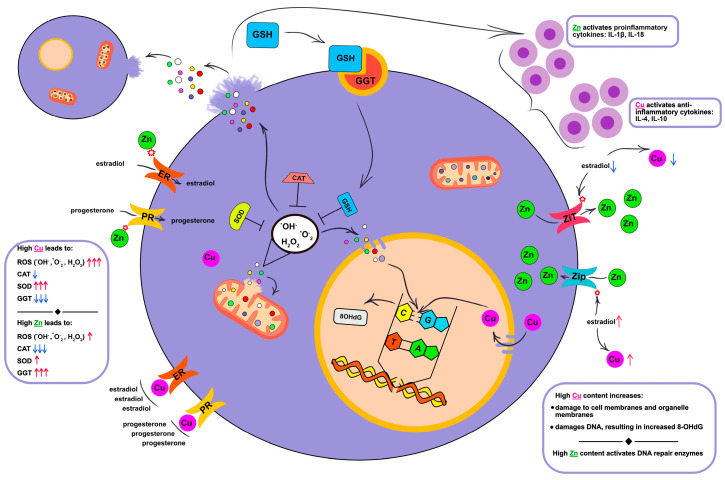
The relationship between copper and zinc metabolism and antioxidant defense enzymes, the immune system, and hormonal levels. ROS—reactive oxygen species; CAT—catalase; SOD—superoxide dismutase; GGT—gamma-glutamyl transferase; GSH—glutathione; ER—estrogen receptor; PR—progesterone receptor; 8OHdG—8-hydroxydeoxyguanosine; ^•^OH^−^—hydroxyl radical; H_2_O_2_—hydrogen peroxide; ^•^O_2_^−^—superoxide anion. Blue arrows indicate a decrease in enzyme activity/decrease in concentration; red arrows indicate an increase in enzyme activity/increase in concentration. The number of arrows shows the intensity of the parameter change.

**Table 1 ijms-26-04784-t001:** Comparison of the saliva composition in breast cancer, fibroadenomas, and the healthy controls.

Indicators	Breast Cancer, *n* = 230	Fibroadenomas, *n* = 92	Healthy Controls, *n* = 59	Kruskal–Wallis Test; *p*-Value
Age, years	60.2 [50.1; 66.5]	52.0 [45.0; 62.0]	53.7 [40.6; 60.3]	5.324; 0.0741
8-OHdG, pg/mL	235.5 [156.3; 636.2]	299.2 [249.5; 347.1]	188.4 [89.1; 309.1]	4.866; 0.0926
CRP, mU/mL	0.176 [0.122; 0.312]	-	0.153 [0.118; 0.212]	0.2528
**Cytokines**				
IL-1β, pg/mL	124.8 [30.84; 305.4]	144.5 [37.00; 310.1]	37.01 [11.78; 106.1]	8.796; 0.0123 *
IL-4, pg/mL	2.49 [1.75; 4.04]	2.79 [1.74; 4.88]	1.61 [1.03; 2.96]	18.12; 0.0001 *
IL-10, pg/mL	4.70 [3.30; 7.15]	4.97 [3.78; 6.38]	2.25 [1.68; 3.48]	44.73; 0.0000 *
IL-18, pg/mL	67.05 [31.13; 132.9]	62.50 [40.90; 134.3]	63.86 [22.50; 141.8]	0.3925; 0.8218
**Hormones**				
Estradiol, nmol/L	3.01 [2.55; 3.72]	-	3.62 [2.63; 5.20]	0.2076
Progesterone, nmol/L	2.47 [2.08; 2.99]	-	2.67 [2.34; 3.08]	0.4281
**Activity of the antioxidant system**				
Catalase, nkat/L	3.77 [2.56; 5.94]	3.64 [2.48; 5.21]	4.58 [3.32; 5.79]	18.46; 0.0001 *
SOD, c.u.	73.7 [34.2; 142.1]	63.2 [34.2; 115.8]	57.9 [31.6; 113.2]	7.223; 0.0270 *
NO, μmol/L	28.3 [18.3; 41.9]	32.7 [18.4; 55.8]	22.8 [13.2; 36.8]	35.74; 0.0000 *
MDA, μmol/L	6.92 [5.56; 8.80]	7.01 [5.90; 8.97]	6.50 [5.73; 7.95]	14.60; 0.0007 *
GGT, U/L	23.3 [20.0; 26.5]	21.5 [18.9; 24.4]	20.4 [17.4; 22.4]	50.92; 0.0000 *
**Amino acids**				
Cys, nmol/L	1.33 [0.59; 2.63]	1.58 [1.10; 2.68]	2.18 [1.04; 3.57]	3.604; 0.1649
Met, nmol/L	4.97 [3.38; 7.12]	6.61 [3.28; 10.35]	4.16 [1.85; 5.30]	7.837; 0.0199 *
His, nmol/L	19.44 [12.76; 30.00]	20.31 [15.67; 36.75]	24.85 [21.25; 29.24]	4.582; 0.1012
Arg, nmol/L	10.22 [5.95; 21.23]	13.78 [6.83; 21.91]	25.23 [20.72; 38.10]	14.55; 0.0007 *

Note: * Differences between the three groups (breast cancer, fibroadenomas, and healthy controls) were statistically significant, *p* < 0.05.

## Data Availability

The raw data supporting the conclusions of this article will be made available by the authors on request.

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
