# Peer review of "Salivary Zinc and Copper Levels Are Differentially Associated with ROS Levels in Breast Cancer Patients"

_ijms, 2025, doi:10.3390/ijms26104784_

Round 1
Reviewer 1 Report
Comments and Suggestions for Authors
The article provides a comprehensive exploration of salivary zinc and copper levels and their relationship to breast cancer phenotypes. The authors elucidate the metabolic pathways involving these metals and their implications for oxidative stress, immune response, and cancer progression. However, integrating additional research would further strengthen the manuscript. Specifically:
-
Incorporate insights from studies such as DOI: 10.3390/antiox13070841 and DOI: 10.3390/antiox13050609. These works delve into the role of metals in carcinogenesis, expanding on the mechanisms through which copper and zinc influence cancer cell behavior, particularly through oxidative stress and redox homeostasis.
-
Emphasize the genetic predispositions linked to breast cancer, such as mutations in the BRCA1 gene, which elevate breast cancer risk up to 70%. Discuss how the interplay between genetic factors and metal metabolism might create synergistic effects, potentially exacerbating carcinogenic processes.
-
Highlight gaps in current research, such as the need for a deeper understanding of how salivary diagnostics could offer non-invasive biomarkers for early detection and differentiation of breast cancer subtypes.
Adding these perspectives would not only enrich the current study but also align it with broader scientific discussions on the intersection of metal metabolism and genetic susceptibility in cancer progression.
Reviewer 2 Report
Comments and Suggestions for Authors
The imbalanced metabolism of copper and zinc can lead to various pathological conditions, including cancer. Zinc supports enzyme activity in immune, redox, and inflammatory processes and DNA regulation, while copper influences cell differentiation, epithelial-mesenchymal transition, and cancer progression. Studying changes in copper and zinc in patients with breast cancer can provide valuable insights into their metabolic features. This study examined the relationship between salivary copper and zinc levels and breast cancer, analyzing their link to oxidative stress, immune function, amino acid metabolism (Cys, His, Met, Arg), hormonal status, and hormone receptor expression. Although these findings are promising, they also raise some concerns/questions and need clarification/explanation.
Major comments
- The authors have mentioned the age range of the patients they recruited for sample collection. Could they provide the mean age of the patients and healthy volunteers?
- Will the authors confirm whether any of the patients or volunteers had co-morbid conditions? Co-morbidities can influence serum levels of metal ions and may impact other malignancies. Additionally, dietary intake of zinc could affect its serum levels. Co-morbid conditions might also influence cytokine levels. The authors should clarify whether these factors were considered.
- The association between serum zinc and copper levels and breast cancer remains unclear. The influence of hormones on zinc and copper metabolism is not fully understood. It would be helpful if the authors presented correlation data between hormone levels and those of zinc and copper, as well as correlations between zinc and copper levels and amino acid levels.
- Iron plays a critical role in breast cancer progression by generating reactive oxygen species (ROS) and inducing oxidative stress. Why was iron not considered in the study?
Other comments:
Line 13: “the changes in copper and zinc in p”- please check
Line 381: From Arg, under the influence of NOS (NO synthase), Cit and NO are formed. what is “cit”?
Round 2
Reviewer 2 Report
Comments and Suggestions for Authors
I would like to thank the authors for their clarification.